# Hybridized Hierarchical Watermarking and Selective Encryption for Social Image Security

**DOI:** 10.3390/e25071031

**Published:** 2023-07-07

**Authors:** Conghuan Ye, Shenglong Tan, Zheng Wang, Binghua Shi, Li Shi

**Affiliations:** 1School of Information Engineering, Hubei University of Economics, Wuhan 430205, Chinawangzheng@hbue.edu.cn (Z.W.); p2pgrid@gmail.com (B.S.); shily0118@163.com (L.S.); 2Hubei Internet Finance Information Engineering Technology Research Center, Hubei University of Economics, Wuhan 430205, China

**Keywords:** social network analysis, social image sharing, image encryption, joint watermarking and encryption, jiugongge game

## Abstract

With the advent of cloud computing and social multimedia communication, more and more social images are being collected on social media platforms, such as Facebook, TikTok, Flirk, and YouTube. The amount of social images produced and disseminated is rapidly increasing. Meanwhile, cloud computing-assisted social media platforms have made social image dissemination more and more efficient. There exists an unstoppable trend of fake/unauthorized social image dissemination. The growth of social image sharing underscores potential security risks for illegal use, such as image forgery, malicious copying, piracy exposure, plagiarism, and misappropriation. Therefore, secure social image dissemination has become urgent and critical on social media platforms. The authors propose a secure scheme for social image dissemination on social media platforms. The main objective is to make a map between the tree structure Haar (TSH) transform and the hierarchical community structure of a social network. First, perform the TSH transform on a social image using social network analysis (SNA). Second, all users in a social media platform are coded using SNA. Third, watermarking and encryption are performed in a compressed domain for protecting social image dissemination. Finally, the encrypted and watermarked contents are delivered to users via a hybrid multicast–unicast scheme. The use of encryption along with watermarking can provide double protection for social image dissemination. The theory analysis and experimental results demonstrate the effectiveness of the proposed scheme.

## 1. Introduction

With the advent of cloud computing and social multimedia communication, more and more social images are being collected on social network platforms, such as Facebook, TikTok, Flirk, and YouTube. The amount of social images produced and disseminated is rapidly increasing. One of the fastest-growing types of social media relates to user-generated social multimedia (audio, video, and images) on social media platforms. There exists an unstoppable trend of the dissemination of fake/unauthorized social images and videos. Meanwhile, cloud computing-assisted social media platforms have made social image dissemination more and more efficient.

The growth of social image collection and dissemination through social media platforms is supported by resource-constrained sensor devices, such as wireless sensor nodes, smartphones, and other network-enabled appliances. Social media is an emerging paradigm applied to content dissemination. It is beyond the ability of traditional media properties. It is often from various sources yet unstructured, such as video and image archives, sensors, scientific applications, surveillance, Internet texts and documents, medical records, and weblogs. The emerging social media platform makes social media sharing very easy. It has become an integral part of society. On social media platforms, social multimedia dissemination becomes more and more popular. The growth of social image sharing underscores potential security risks for illegal use, such as image forgery, malicious copying, piracy exposure, plagiarism, and misappropriation. Preserving privacy in publishing social images becomes an important concern. More importantly, a social image has to be highly secure and establish privacy for dissemination on social media platforms.

To ensure the security of social images, secure social image dissemination on social media platforms is becoming increasingly urgent. To protect the privacy of social images on social media platforms, techniques are needed to ensure social image security in terms of its source identity, content integrity, privacy, trust, and anonymity. There are many measures for protecting the privacy of a social image, such as encryption and watermarking. Encryption is a method with which the privacy of a social image can be ensured and any unauthorized access can be prevented [1,2,3,4,5,6]. However, once users of social media platforms receive and decrypt the social image, it could be copied and re-disseminated arbitrarily. The social image is not protected in this case. Thus, social image redistribution should be deterred by extra protection schemes. Watermarking is another possible way to protect a social image further [7]. Watermarking protocols have been proposed to protect multimedia content in [8,9]. In the scenario of social image dissemination on social media platforms, the personal information of a user should be embedded into the social media to trace illegal use [10]. Because plain social media is easily attacked during personal information insertion, a watermark embedded in the encrypted media is a good solution. The encrypted media is protected by cryptosystems [11,12,13,14,15]. However, watermarking cannot trace somebody who redistributed the copy. In the practical application scenario, as a special case, digital fingerprinting can deter the redistribution of content on social media platforms [16,17,18].

A social image is easy to be processed by copying and disseminating without quality distortion. Therefore, it is becoming more and more urgent to resist unauthorized manipulation. Privacy protection on a social image is imperative in the whole life cycle. Because fundamental roles are quite distinct, encryption and watermarking [17,19,20,21] have been researched independently most of the time. Multimedia encryption and watermarking are different research directions, and all previous schemes are mainly concentrated on the spatial domain or the transform domain. Neither multimedia encryption nor watermarking can provide double-layer protection for multimedia dissemination.

During the past few years, some pioneering research has been proposed in this field. The advancement of a combination between watermarking and cryptography is fostered [22,23,24,25,26,27]. For information hiding, Bianchi et al. [28] have investigated how to implement the discrete Fourier transform in the encrypted domain. Novel solutions will be provided entirely for security and privacy preservation on social images. A novel architecture for joint fingerprinting and decryption has been proposed by Kundur et al. [29]. A better compromise between security and practicality can be ensured by the marriage structure. Lian SG et al. [30] have proposed a joint scheme of fingerprinting, encryption, and encoding for multimedia content dissemination. All the mentioned schemes have been conducted in the spatial/transform domain. They can meet the part requirements of secure multimedia content dissemination [31]. Some schemes propose the convergence of encryption and watermarking, and they are now facilitating social media privacy and security studies [32]. However, none of them apply to social image dissemination in the compressed domain for the resource-constrained social media platforms.

Multimedia dissemination on social media platforms may cause some security and privacy concerns. For secure social multimedia distribution, content confidentiality and redistribution tracing are needed to be protected. Encryption and fingerprinting can resolve them, respectively. However, the relationship and scale of users on social media platforms are not considered by the traditional watermarking methods. The joint encryption and traditional watermarking scheme cannot be applied to secure multimedia sharing on social media platforms. Furthermore, existing joint watermarking and encryption methods for social image dissemination on social media platforms face challenges in big data and distribution efficiency problems. The traditional joint watermarking and encryption schemes do not concern the tremendous scale of users on social media platforms and the big data problem that may be caused by fingerprinted social image sharing. They do not even bother to resolve the big data problem in the encrypted–compressed domain. To overcome those challenges, this research proposes a scalable social image security scheme in the compressed–encrypted domain for content dissemination.

As a general wavelet transform, the TSH transform can extract different kinds of information from the media. Encryption and watermarking are therefore very likely to be conducted in the TSH transform domain [33]. In this study, the first joint encryption and fingerprinting method is proposed in the TSH transform domain of encrypted multimedia content. Using social network analysis (SNA) [34], a map between social relationships of users on social media platforms and the TSH wavelet transform is proposed to deal with the issues of secure dissemination and redistribution tracing. The proposed marriage structure offers a discussion of how to use the TSH transform to realize secure multimedia dissemination on social media platforms.

The classical discrete wavelet transform (DWT) is a special case of the TSH transform [35]. The authors address the issue of secure multimedia dissemination using watermarking/encryption in the TSH domain. This paper will provide an SNA method for secure multimedia dissemination for social media platforms. First, a fingerprint code is designed by the dendrogram of the social network structure, which is also used to conduct the TSH transform. Second, a watermarked–encrypted method is proposed in the compressed domain. Finally, the secure media content is disseminated with hybrid multicast–unicast mode by SNA. With the proposed scheme, a privacy-preserving and secure multimedia dissemination method can be applied on social media platforms. The distribution efficiency, privacy preserving, and redistribution tracing can be ensured [32].

The remainder of this work is organized as follows. Some basic theories are introduced in Section 2. In Section 3, the proposed scheme is presented. Section 4 analyzes the performance of the scheme and its security. The conclusion and future works are in Section 5.

## 2. Preliminaries

This section discusses the basic theories involved in the recommended image protection scheme. Social network analysis decides the fingerprint code structure and the TSH wavelet transform. SHA-3 and chaotic maps are used to encrypt the image. Joint watermarking and encryption in the compressed domain will not only protect the image for a high level of security for social image sharing.

### 2.1. SNA

In this research, two users on a social media platform have a social relationship if they communicate with each other on the platform. A graph is used to represent the social relationships of users on the social media platform. Graph G=(V,E) with *M* nodes can be described by an adjacency matrix *A*. If the link exists, its corresponding edge $a_{ij}$ (*i*, *j* = 1, …, *M*) is equal to 1, or it will be zero. For graph G=(V,E), the main objective is to identify the community structure with a partition $P=\{U_{1},U_{2},\;\;\ldots \;\;\;,U_{c}\}$, and *c* is the community number. Figure 1 shows four communities.

### 2.2. The TSH Transform

A multimedia social network is modeled by the graph G=(V,E). *V* is the node set, which represents users in the multimedia social network. *E* is the edge set, which shows the relationship between two users. Adjacency matrix *A* can be used to represent the graph, and $a_{ij}$ (*i*, *j* = 1, …, *M*) is 1 if the related edge exists; otherwise, it is zero. For G=(V,E), a partition $P=\{S_{1},S_{2},\;\;\ldots \;\;\;,S_{c}\}$ is the communities of the multimedia social network. The number *c* represents the community number of a multimedia social network.

On the other hand, the TSH transform includes non-flat TSH functions and flat TSH functions. The binary interval splitting tree (BIST) can help to define TSH functions [35]. At the same time, BIST can also show the hierarchical characteristics of TSH functions [33].

To an interval I=[1,N], where $N=2^{L}$, *I* is split by $I_{0}^{H}=\left[1,2^{-1}N\right]$ and $I_{1}^{H}=\left[2^{-1}N+1,N\right]$ for constructing the function basis. Then, the TSH wavelet function can be defined as follows:(1)$$I_{0}^{H}=\left[1,2^{-1}N\right]$$
(2)$$\begin{array}{c} \Psi^{H}\left(t\right)=\left\{\begin{array}{c} \frac{1}{\sqrt{N}}\;\;\;\;\;\;\;\;\;\;\;\;\;\;\;\;\;\;\;\;\;\;\;\;for\;\;\;t\in I_{0}^{H}\\ -\frac{1}{\sqrt{N}}\;\;\;\;\;\;\;\;\;\;\;\;\;\;\;for\;\;\;t\in I_{1}^{H}\\ 0\;\;\;\;\;\;\;\;\;\;\;\;\;\;\;\;\;\;\;\;\;\;\;\;\;\;\;\;\;\;\;\;\;\;otherwise \end{array}\right. \end{array}$$
where $\left|\right|\Psi^{H}\left(t\right)\left|\right|=1$ and $\Psi^{H}\left(t\right)$ is orthogonal, and ||•|| is the norm. For the general TSH transform, *I* is partitioned by $I_{0}^{TSH}=\left[1,v_{0}\right]$ and $I_{1}^{TSH}=\left[v_{0}+1,N\right]$, $1\leq v_{0}<N$. Function $\Psi^{TSH}\left(t\right)$ is
(3)$$\begin{array}{c} \Psi^{TSH}\left(t\right)=\left\{\begin{array}{c} \sqrt{\frac{v_{1}}{v_{0}N}}\;\;\;\;\;\;\;\;\;\;\;\;\;\;\;\;\;\;\;\;\;\;\;\;for\;\;\;t\in I_{0}^{TSH},\\ -\sqrt{\frac{v_{0}}{v_{1}N}}\;\;\;\;\;\;\;\;\;\;\;\;\;\;\;for\;\;\;t\in I_{1}^{TSH},\\ 0\;\;\;\;\;\;\;\;\;\;\;\;\;\;\;\;\;\;\;\;\;\;\;\;\;\;\;\;\;\;\;\;\;\;otherwise. \end{array}\right. \end{array}$$
where $v_{1}=N-v_{0}$. This construction can be iterated by splitting $I_{0}^{TSH}$ into $I_{00}^{TSH}$ and $I_{01}^{TSH}$ and $I_{1}^{TSH}$ into $I_{10}^{TSH}$ and $I_{11}^{TSH}$. All *N* functions $\Psi^{TSH}\left(t\right)\left(1,N,v_{0}\right)$ are TSH functions which are orthogonal.

### 2.3. Secure Hash Algorithm (SHA-3)

A cryptographic hash function can map any length of message to a fixed length. It can be used for authenticated encryption and pseudo random number generation. Usually, a cryptographic hash function has the sensitivity property of the input message. However, the collisions of the Message-Digest algorithm (MD5) have been found [36,37,38]. Therefore, with higher security requirements, MD5 should be avoided in certain domains.

SHA-3, also known as Keccak, is a widely used cryptographic hash function [39]. The design of SHA-3 was based on permutation functions. For any type of message as input, the SHA-3 function can generate different length hash values through computing the given message. They have fixed 224-, 256-, 384-, and 512-bit hash values. In this paper, the 256-bit hash value will be used as output in the proposed algorithm. The hash value can ensure the integrity and consistency of information transmission. In addition, SHA-3 is very sensitive to the input message. Even if there is only a tiny bit of change between two input sequences, the returned hash value will be totally different. The time performance of the SHA-3 function is superior because of its bit-level operations. Because it has fast computation capability and is very sensitive to original input content, it will be used to design keys of medical image encryption in this paper.

### 2.4. Chaotic Maps

Multimedia cryptosystems using chaotic maps have gained large amounts of concern recently. They have the most attractive features, which include sensitivity to initial values, the control parameters, random trajectory, the outspreading of orbits diffusing to the whole space, etc. Even the simplest chaotic system can show perfect ergodicity and randomness. Therefore, the proposed algorithm mainly uses MD5 and the following chaotic maps to encrypt images in the TSH wavelet transform domain.

The logistic map [40] is a well-known continuous dynamical system nowadays. Although they are very simple, logic maps have complex dynamic behaviors like other maps. The one-dimensional logistic map can be described as follows:(4)$$t_{n+1}=ut_{n}\left(1-t_{n}\right)$$
where $t_{n}$∈ (0, 1), and n is a natural number. *u*∈ [0, 4], and if 3.56994<u≤4, the one-dimensional logistic system will in a chaotic state.

The piecewise linear chaotic map (PWLCM) [41] can show better complex dynamic behaviors than the logistic map. It is mainly composed of piecewise linear functions. It can be described as follows:(5)$$z_{n+1}=F\left(z_{n},\eta \right)=\left\{\begin{array}{c} z_{n}/\eta ,\;\;\;\;\;\;\;\;\;\;\;\;\;\;\;\;\;\;\;\;\;\;\;\;\;\;\;\;\;\;\;\;\;\;\;\;\;\;\;\;\;\;\;\;\;\;\;\;\;\;\;\;\;\;\;\;\;\;\;\;\;\;0\leq z_{n}<\eta \\ \left(z_{n}-\eta \right)/\left(0.5-\eta \right),\;\;\;\;\;\;\eta \leq z_{n}<0.5\\ 0,\;\;\;\;\;\;\;\;\;\;\;\;\;\;\;\;\;\;\;\;\;\;\;\;\;\;\;\;\;\;\;\;\;\;\;\;\;\;\;\;\;\;\;\;\;\;\;\;\;\;\;\;\;\;\;\;\;\;\;\;\;\;\;\;\;\;\;z_{n}=0.5\\ F\left(1-z_{n},\eta \right),\;\;\;\;\;\;\;\;\;\;\;\;\;\;\;\;\;\;\;\;\;\;\;\;\;\;\;\;\;\;\;\;\;\;\;\;0.5\leq z_{n}<1 \end{array}\right.$$
where $z_{n}\in \left(0,1\right)$, and *n* is a natural number. If η∈(0,0.5), the PWLCM map will be in a complex chaotic state.

### 2.5. Joint Watermarking and Encryption in the Compressed Domain

Joint social fingerprinting and encryption can not only provide double-level privacy protection for content dissemination on social media platforms but also bring an efficient content distribution scheme to avoid big data effects. For the field of security, digit watermarking/fingerprinting can be regarded as another cryptographic method for content protection on social media platforms.

For integrated fingerprinting and encryption for protection, there exist two different research directions. The first one incorporates fingerprinting into decryption algorithms. Another aims at combining fingerprinting and encryption, which is called the joint fingerprinting and encryption (JFE) scheme [42]. Multimedia encryption schemes can be classified as full encryption and selective encryption. As for the former, all contents are selected to encrypt. All content can be protected. Any information about the original content is not leaked. On the contrary, the latter only selects the most crucial part of the original content to encrypt. Although there are marriages between the digital fingerprinting and encryption of these existing methods for content sharing on social media platforms, certain uncertainty about the length of the fingerprint code and distribution performance has not been discussed carefully. Thus, their methods are not appropriate for secure content dissemination on social media platforms because of the big data problems.

How to use SNA to embed fingerprints in the compressed–encrypted contents and how to make the content dissemination with privacy protection on social media platforms is not deeply researched in the existing works either. Undoubtedly, safeguarding privacy and security of personal information on social media platforms is still in its infancy. On social media platforms, practical multimedia contents are stored and transmitted in the compressed format. The fingerprinting and encryption should be implemented in the compressed content to avoid the process of fully decoding and encoding. Understanding the inherent characteristics of the JPEG and the JPEG2000 may play a useful role in digital image forensics. To address these issues, we present a novel JFE scheme based on the jiugongge game using SNA in the TSH wavelet transform domain for persistent privacy protection. In our proposed scheme, the jiugongge game [5] is a kind of sudoku game [43,44], which is an interesting platform for digital image scrambling by offering the benefit of encryption computation before JPEG2000 compression. To increase the encryption effect further, the probabilistic homomorphic cryptosystem is used in the JPEG2000-compressed byte stream.

### 2.6. Main Contribution

The proposed sharing method in the compressed–encrypted domain using SNA is mainly to research the security and privacy issues on social media platforms. The main contributions of the scheme are as follows:

(1) The privacy protection method proposes how to use the JFE for secure dissemination in the compressed–encrypted domain.

(2) The proposed scheme uses content self-adaptively; the user’s social fingerprint code is produced according to the original multimedia content.

(3) The proposed scheme can provide technology for double encryption in the compressed domain before content coding and after the JPEG2000 coding stage.

(4) The proposed scheme can avoid big data problems on social media platforms to the utmost extent with the JFE in the compressed domain.

## 3. Proposed Scheme

Secure social image dissemination is becoming increasingly urgent on social media platforms. A manner for content security and privacy preserving should be ensured during the process of multimedia content dissemination. To protect social media content, there should exist two properties, such as confidentiality and redistribution tracing. For realizing secure dissemination of content on social media platforms, encryption transforms plaintext into an unintelligible and enciphered form. The enciphered content must ideally “appear” random. Without the decryption keys, it is difficult to estimate the original content from the enciphered content. Digital fingerprinting can resolve the redistribution tracing issue. Fingerprinting can use watermarking techniques to embed fingerprints into social media content. Usually, in order to retain the perceptual security of the protected content, the host signal should be changed subtly. Because fundamental roles are quite distinct, encryption and watermarking have been researched independently most of the time. Multimedia encryption and watermarking are different research directions for multimedia security. However, with only multimedia encryption methods, no protection will be provided if the encrypted content is decrypted. Any user can redistribute the decrypted contents to others without any cost.

The marriage of fingerprinting and encryption can facilitate security and privacy studies on social media platforms. Thus, encryption and fingerprinting can be regarded as an “a priori” protection method and an “a posteriori” control method, respectively. The authors mainly focus on how to combine them to realize secure dissemination of a social image on social media platforms. The highest-level approximation component in the TSH domain is selected to embed the users’ personal fingerprint codeword. The other components are chosen to embed the community fingerprint code segments of the users. In Figure 2, the proposed marriage scheme is shown. First, the original social image is decomposed by the TSH wavelet transform. The approximation coefficient is selected for permutation with jiugongge block maps. Second, with a probabilistic homomorphic cryptosystem, the compressed contents are encrypted totally. Third, fingerprint information is embedded into the corresponding encrypted coefficients. Finally, the compressed–encrypted–fingerprinted content is distributed on social media platforms.

### 3.1. Fingerprint Encoding Using SNA

On a social media platform, all the users from the platform are grouped into overlapping and hierarchical communities. With this dendrogram, the fingerprint code can be designed through the tree-based structure mode which can reduce the code length. Users who have a direct social relationship belong to the same community. They have the same code structures. Their multilevel community code segments are the same.

Figure 1 is a community structure of users from a social media platform. All users are grouped into four different communities. The Boneh D., Shaw J. (BS) code [45] is used to design the multilevel community code. With the Tardos scheme [46], every user’s personal fingerprint code can be designed. Therefore, the unique identification information of users can be encoded through concatenation of a multilevel community code and the user personal fingerprint code.

In the proposed fingerprints encoding design, users who are from the same community own the same multilevel community code segment. Therefore, their fingerprinted–encrypted–compressed media contents will have the same part, which can be disseminated to the users via the multicast mode. Good fingerprint code structure plays an essential role in cost-efficient content dissemination schemes.

### 3.2. Discontinuity Point Vector

Each splitting scheme defines a different basis set of the TSH wavelet transform. For a given *N*, each TSH basis is univocally defined by a Discontinuity Point Vector (DPV) [33], and the DPV defines the splitting scheme. Each Haar function has two non-zero values and one break point, which occurs at a dyadic location. To define TSH functions, a type of labeled binary tree, called binary interval splitting tree (BIST), is employed [35]. In addition, BIST also serves the purpose of illustrating hierarchical dependencies among TSH functions [33]. Given the discrete interval I=[0,N] with $N=2^{L}$, we split the interval into two halves $I_{0}^{H}=\left[1,2^{-1}N\right]$ and $I_{1}^{H}=\left[2^{-1}N+1,N\right]$, where *H* represents Haar.

In this paper, we determine the DPV based on the structure of the fingerprint code. For example, if the number of layers of the community structure is n+1, so the number of segments of fingerprint code is n+1, then the length of the DPV is *n*, and interval *I* will be split into n+1 intervals, and the sizes of these intervals are decided by the length of the fingerprint code, where $v_{i}\left(i=0,1,\ldots ,n-1\right)$ is the splitting node. Figure 3 shows an example of BIST; it serves the purpose of illustrating hierarchical dependencies among TSH functions. For example, from a given interval as shown in Figure 3, one can construct the corresponding complete binary tree and the tag binary tree. It can produce a DPV, with which the tree structure Haar wavelet transform can perform.

In order to make a one-to-one map between the social fingerprint code and the TSH transform, the DPV is determined with SNA. For a given community structure of users from a social media platform in Figure 1, assume the number of community layers is n+1, the DPV’s length is *n*, and n+1 intervals will partition the interval *I*. The length of every level community code can decide the size of the corresponding interval. The related splitting node is $v_{i}\left(i=0,1,\ldots ,n-1\right)$. Assume $a_{0}^{TSH}=1$, $b_{0}^{TSH}=v_{0}$, $a_{1}^{TSH}=v_{0}+1$, $b_{1}^{TSH}=N$. The TSH basis function $\Psi^{TSH}\left(t\right)$ can be represented as follows:
(6)$$\Psi^{TSH}\left(t\right)=\left\{\begin{array}{c} \sqrt{\frac{N-v_{0}}{v_{0}N}}\;\;\;\;\;\;\;\;\;\;\;\;\;\;\;\;\;\;\;\;\;\;\;\;for\;\;\;t\in I_{0}^{TSH},\\ -\sqrt{\frac{v_{0}}{N(N-v_{0})}}\;\;\;\;\;\;\;\;\;\;\;\;\;\;\;for\;\;\;t\in I_{1}^{TSH},\\ 0\;\;\;\;\;\;\;\;\;\;\;\;\;\;\;\;\;\;\;\;\;\;\;\;\;\;\;\;\;\;\;\;\;\;otherwise. \end{array}\right.$$
where $v_{1}=N-v_{0}$. This construction can be iterated by splitting $I_{0}^{TSH}$ into $I_{00}^{TSH}$ and $I_{01}^{TSH}$ and $I_{1}^{TSH}$ into $I_{10}^{TSH}$ and $I_{11}^{TSH}$, respectively, and so on. The set of *N* functions $\Psi^{TSH}\left(t\right)\left(1,N,v_{0}\right)$ is called TSH functions. It can be proved that the set of TSH functions is a set of orthogonal functions.

For a node $\left(\alpha_{1},\alpha_{2},\ldots ,\alpha_{k}\right)$, the partition mode is the same, in other words, a section of length $L_{\alpha_{1},\alpha_{2},\ldots ,\alpha_{k}}^{TSH}=b_{\alpha_{1},\alpha_{2},\ldots ,\alpha_{k}}^{TSH}-a_{\alpha_{1},\alpha_{2},\ldots ,\alpha_{k}}^{TSH}+1$, and the interval $I_{\alpha_{1},\alpha_{2},\ldots ,\alpha_{k}}^{TSH}=\left[a_{\alpha_{1},\alpha_{2},\ldots ,\alpha_{k}}^{TSH},b_{\alpha_{1},\alpha_{2},\ldots ,\alpha_{k}}^{TSH}\right]$, $\alpha_{i}=\left\{0,1\right\}$ can be partitioned as follows:(7)$$\begin{array}{c} I_{{}_{\alpha_{1},\alpha_{2},\ldots ,\alpha_{k}}}^{TSH}=\left|I_{\alpha_{1},\alpha_{2},\ldots ,\alpha_{k},0}\right|\frac{v_{\left(\alpha_{1},\alpha_{2},\ldots ,\alpha_{k},1\right)}}{v_{\left(\alpha_{1},\alpha_{2},\ldots ,\alpha_{k}\right)}v_{\left(\alpha_{1},\alpha_{2},\ldots ,\alpha_{k},0\right)}}\;\\ \;\;\;\;\;\;\;\;\;\;\;\;\;\;\;\;\;\;\;\;\;\;\;\;\;\;\;\;\;\;\;\;\;\;\;\;\;\;\;\;\;\;\;\;\;\;+|I_{\alpha_{1},\alpha_{2},\ldots ,\alpha_{k},1}|\frac{v_{\left(\alpha_{1},\alpha_{2},\ldots ,\alpha_{k},0\right)}}{v_{\left(\alpha_{1},\alpha_{2},\ldots ,\alpha_{k}\right)}v_{\left(\alpha_{1},\alpha_{2},\ldots ,\alpha_{k},1\right)}} \end{array}$$

### 3.3. The Jiugongge Permutation System

The jiugongge game [5] is a kind of sudoku game [43,44], which rearranges 9 numbers with some rules. The rule of the jiugongge game is to separately put numbers 1∼9 into a 3-row, 3-column table cell. In the end, the sum of the three numbers on the vertical, horizontal, and diagonal lines is equal to 15.

In this research, the jiugongge game is used to confuse wavelet coefficients. Each subband in the TSH wavelet domain can be divided into some blocks. The default size of these blocks is 8×8. The size can be adjusted to suit particular security needs. Each 3×3 grid block is grouped as a jiugongge system, which is numbered 1 to 9, respectively, as shown in Figure 4a. In Figure 4a, based on the jiugongge game, the blocks are rearranged following jiugongge rules as shown in Figure 4c: the numbered block 1 is put into block 2 of Figure 4b, block 2 in Figure 4a to block 9 of Figure 4b, and the other blocks’ movements are shown in Figure 4. Once all blocks in Figure 4a are moved into Figure 4c, the proposed wavelet coefficient block permutation based on the jiugongge game is completed as shown in Figure 4c. Because the location of block 5 is not changed, only 8 blocks are needed to move in the jiugongge game system. To make the wavelet coefficient block permutation in its entirety, each subband of the TSH domain follows the jiugongge rules to realize the scrambling process.

### 3.4. JPEG2000 Code Based on the TSH Transform

As a generic wavelet transform, the TSH wavelet transform is widely applied in the field of signal processing. Compared with the DWT transform, the TSH wavelet allows each Haar-like function to break at nondyadic locations. Because of the one-to-one map relationship between the TSH transform and social fingerprint code structure, the fingerprint code segments are very suitable to be embedded into the coefficients of the TSH transform. Through the TSH transform of the encrypted content, the fingerprint codeword can be embedded into the encrypted–compressed domain. With the additive homomorphic operation, every level fingerprint code segment is embedded in the encrypted–compressed domain in the corresponding subbands. The corresponding social fingerprint code segment is embedded into the selected encrypted coefficient. The unique user code is embedded into the highest-level approximation in the TSH domain, and the multilevel community code segments are embedded into the other coefficients.

Given a social image, perform the discrete TSH wavelet transform based on the DPV, which can produce a multi-resolution image. Then, compress the multi-resolution social image with the JPEG2000 standard. The whole compression process is divided into three different stages. First, the wavelet coefficients are quantized. Second, the quantized coefficients are decomposed into bit planes, which are partitioned into different code-blocks. Finally, at embedded block coding with optimized truncation (EBCOT), through multiple passes, each code-block is encoded independently to produce the compressed byte stream which is grouped into different wavelet packets. In this case, from different bit planes of the wavelet coefficient, different wavelet packets are chosen to encrypt and watermark independently with the homomorphic cryptosystem directly.

### 3.5. Probabilistic Homomorphic Cryptosystem

The proposed probabilistic homomorphic cryptosystem is based on a public key cryptosystem with semantic security [47,48].

Two plaintexts, *x* and *y*, two numbers, $r_{1}$ and $r_{2}$, where E(·) is encryption operation with public key. D(·) denotes decryption with private key. The product of two large primes modulus is *z*. The proposed probabilistic homomorphic cryptosystem mainly uses the following calculation rules.
(8)$$D(E\left(x,r_{1}\right)•E\left(y,r_{2}\right))=x+y\;\;\;\mathrm{mod}\;\;\;z$$
(9)$$D\left(E{\left(x,r\right)}^{k}\right)=D(E\left(kx,r\right)=kx\;\;\;\mathrm{mod}\;\;\;z$$

Given a chaotic sequence $y_{i}$, the 8-bit random key stream $r_{i}$ is produced according to the following formula.
(10)$$r_{i}=mod(\left(floor\left(y_{i}\times 2^{48}\right),256\right),i=0,1,\ldots$$
and then the homomorphic encryption is processed byte by byte to obtain the ciphered sequence:(11)$$C_{i}=E\left(m_{i},r_{i}\right),i=0,1,\ldots$$

The proposed content encryption scheme is mainly based on an additive homomorphic cryptosystem. The homomorphic properties of the Paillier cryptosystem are discussed above. Compared with a symmetrical cryptosystem, a homomorphic cryptosystem as an asymmetric cryptosystem, whose security is better and allows to implement arithmetic operations on the ciphertext, is more suitable for three-party data processing [49]. In addition, the security of homomorphic encryption is assured in [50].

Furthermore, to protect content forever, watermarking with fingerprint information embedded into the encrypted sequence can enhance the resistance to content redistribution. For fingerprint detection in the encrypted domain, the fingerprints are embedded into the ciphered sequence with a Quantization Index Modulation watermarking technique. The unique personal code segment is embedded into the highest-level approximation part in the TSH domain, and the multilevel community code segments are embedded into the other coefficient.

### 3.6. Fingerprint Embedding

To deter illegal redistribution, an encrypted social image should be processed with digital watermarking which is mainly used for fingerprint embedding. Fingerprints are embedded in the compressed and encrypted domain. The multilevel fingerprint code segments are embedded into a compressed and encrypted image. In this research, blind watermarking is applied to multilevel fingerprint code segment embedding. Considering the one-to-one map relationship between multilevel code segments and the TSH transform, the fingerprints are embedded in the TSH domain.

$N_{u}$ is the number of users belonging to a social media platform. $EX^{{}^{I}}=\left(ex_{1},ex_{2},\ldots ,ex_{L}^{I}\right)$ is the encrypted byte stream from the approximation subband for user code segment embedding. Vector $EX^{O}=\left(ex_{1},ex_{2},\ldots ,ex_{L}^{O}\right)$ is the encrypted byte stream in all horizontal and vertical subbands to embed multilevel community code segments. The length of the user code segment and the multilevel community code segment is $EX^{u}$ and $EX^{c}$, respectively, and the user code segment hiding scheme is described as follows:(12)$$EFX_{k}=EX_{k}^{u}+\alpha *F_{k},k=1,2,\ldots ,N_{u}$$
where α is a factor for embedding strength, $F_{k}$ is the fingerprint code segment of the user *k*. The embedding method of the multilevel community code segment is the same as the user code segment. All code segments are embedded into the corresponding encrypted byte stream in parallel because of the independence of code segments and different subbands.

### 3.7. The Proposed Encryption Scheme

Multimedia encryption often transforms the original content into an unintelligible form. It is now well-known that chaotic maps are used for multimedia encryption. The main advantage of chaotic maps lies in that a chaotic signal looks like noise. Only the legal users who have the correct key can decrypt the encrypted content successfully. In the following, we will present JPEG2000 image encryption and decryption. The input is the original social image. The output is the encrypted–compressed–fingerprinted image. It must look like noise to make the estimation of the original social image from the encrypted image computationally difficult without the correct key.

The proposed encryption scheme is based on partial encryption, which means a smaller subset of the important content in the TSH wavelet transform domain is chosen to encrypt for lower computation complexity. The jiugongge game with a chaotic map can develop chaotic permutations with simple rules, which makes the jiugongge game an interesting platform for image confusion. We are interested to use the chaotic jiugongge game to accomplish rapid cryptography. The proposed JPEG2000 image fingerprinting and encryption method is based on the jiugongge game with the probabilistic homomorphic cryptosystem in the TSH wavelet transform domain. The scheme is shown in Figure 2. The encryption is mainly based on permutation with the jiugongge game before the JPEG2000-compressed process and the probabilistic homomorphic encryption after the JPEG2000-compressed process. The proposed encryption scheme is composed of the following steps:

Step 1: Given a social image, turn this image into a one-dimensional vector *I*, obtain a social fingerprint vector $S_{f}$ from *I*. $S_{f}$ is permuted randomly to obtain $S_{f}^{P}$. The initial values and control parameters are generated from the permuted vector $S_{f}^{P}$ using SHA-3, which is a cryptographic hash function. The SHA-3 hash result of $S_{f}^{P}$ is a 256-bit value. *V* is a 128-bit value, which is chosen from the SHA-3 hash result value, and is segmented into eight 16-bit parts $V_{1}$, $V_{2}$, …, $V_{8}$, and turn the values of these parts into decimal numbers. The initial values $x_{0}$, $y_{0}$ and control parameters *u*, η are computed from those decimal numbers. All of them are regarded as the secret keys in the proposed encryption algorithm. The secret keys include: (1) control parameters: η (PWLCM system) and *u* (logistic map); and (2) initial values: $y_{0}$ (PWLCM system) and $x_{0}$ (logistic map).
(13)$$x_{0}=\frac{V_{1}^{Th}}{2^{16}}$$
(14)$$y_{0}=\frac{V_{2}^{Th}}{2^{16}}$$
(15)$$u=3.57+\frac{V_{5}^{Th}}{2^{16}}\times 0.43$$
(16)$$\eta =\frac{V_{6}^{Th}}{2^{17}}$$

Step 2: For the social image *I*, we calculate the one-level TSH wavelet transform coefficient matrix of *I* according to the social fingerprint code. This process can boost the efficiency of fingerprinted content distribution. The LL subband of the TSH wavelet transform is a down-sampled image of *I*. Then, perform two-level TSH wavelet decomposition.

Step 3: Generate a random sequence ($x_{1}x_{2}\cdot \cdot \cdot x_{M\times N}$) with a logistic map. The initial value $x_{0}$ and control parameter *u* are given in advance as keys. A two-dimensional grid of cells $G^{0}$ is created from the random sequence, in which every element is mapped to the corresponding coefficient block in the TSH wavelet transform domain. Then, the jiugongge game is applied to the two-dimensional grid for element permutation, and the coefficient block is permuted accordingly.

Step 4: The permuted content is compressed by the JPEG2000 standard. To protect the permuted content further, probabilistic homomorphic encryption with the PWLCM chaotic map and fingerprints embedding can enhance the security for resistance to attack. The PWLCM map is used to generate random sequences $FP_{M\times N}^{J^{K}}=\left\{fp_{1}^{J^{K}},fp_{2}^{J^{K}},\ldots ,fp_{{}_{M\times N}}^{J^{K}}\right\}$. Then, we can obtain the compressed and encrypted sequence $CP_{M\times N}^{J^{K}}=\left\{cp_{1}^{J^{K}},cp_{2}^{J^{K}},\ldots ,cp_{{}_{M\times N}}^{J^{K}}\right\}$ $cp_{i}$ = ceiling($fp_{i}$ ), which is one-to-one correspondent with the JPEG2000-compressed bytes sequence in the TSH domain, where *J* is the decomposition level, k={LL}.

Step 5: Encrypt all bytes of chosen subbands with probabilistic homomorphic encryption. After all fingerprint code segments are embedded into the encrypted content, the scrambled and fingerprinted social image can be distributed on the social media platform according to user requirements.

### 3.8. Social Image Distribution Scheme

Once the JPEG2000 image is fingerprinted and encrypted, it is ready to be distributed to the friends of the owner on the social media platform. Fingerprint embedding will produce different copies for every user, which must be transmitted using the unicast method. However, the multicast method is more efficient to transmit multimedia content than the broadcast or unicast schemes. For secure multimedia sharing on social media platforms, there are a huge amount of different fingerprinted copies to deliver to a large number of users.

Therefore, it is essential to disseminate fingerprinted and encrypted copies to users efficiently on social media platforms. In this paper, we propose a bandwidth-efficient content dissemination method with the combination of unicast and multicast as shown in Figure 5.

The fingerprint code segments generation method described in this paper can be used to generate a unique user code segment and a common multilevel community code segment using SNA. Figure 1 depicts the relationship between the unique user fingerprint code segment and the shareable multilevel community fingerprint code segment. At the same time, a social image in the TSH wavelet transform domain consists of multi-resolution subbands.

Different subbands can be used to embed different code segments. The LL subband is used to embed the unique user code segment. The fingerprinted LL subband is regarded as principal content. Those shareable multilevel code segments together can be treated as a fingerprint code of a specific community, which is called a community code. Users who belong to the specific community own the same community code segments, which are embedded into the middle LH and HL subbands. The only difference is the principal content with the unique user code segment will be individually disseminated to users as Figure 5 shows. All the LH, HL, and HH subbands with shareable community code segments will be distributed to users with the multicast method. They are regarded as supplementary content. Therefore, the fingerprinted and encrypted content is decomposed into principal content and supplementary content. The principal content with the unique user fingerprint code segment will then deliver to each user using the traditional unicast mode on the social media platform, while the supplementary content with shareable multilevel community code segments will be freely distributed in the community that the users belong to. The social image dissemination procedure is illustrated in Figure 5. The purpose of involving shareable community fingerprint code segments is to enhance the efficiency of content dissemination. The shareable community fingerprint code segments assist to provide a hint about the community to which the social image was redistributed.

According to the proposed JFE scheme for securing content dissemination on social media platforms, the JFE process is performed in the TSH wavelet transform domain. In this method, the keys aim at encrypting and decrypting the social image. They are firstly produced from the fingerprint code. Next, multimedia contents are then fingerprinted and encrypted in the TSH wavelet transform domain. Then, they are distributed to users. A user deploys the keys to decrypt the received principal and supplementary contents. Once all the fingerprinted and encrypted multimedia contents are received, the TSH wavelet transform process is used to obtain the original social media content as follows:(17)$$\begin{array}{cc} w_{LL}\left(J\right) & =\langle G•LL_{J}\rangle \\ w_{LH}\left(j\right) & =\langle G•LH_{j}\rangle ,j=1,2,\ldots ,J\\ w_{HL}\left(j\right) & =\langle G•HL_{j}\rangle ,j=1,2,\ldots ,J\\ w_{HH}\left(j\right) & =\langle G•HH_{j}\rangle ,j=1,2,\ldots ,J \end{array}$$
(18)$$\begin{array}{cc} G= & w_{LL}\left(J\right)LL_{J}+\sum_{j=1}^{J}w_{LH}\left(j\right)LH_{j}+\sum_{j=1}^{J}w_{HL}\left(j\right)HL_{j}\\ \; & +\sum_{j=1}^{J}w_{HH}\left(j\right)HH_{j}.\;\;\;j=1,\ldots ,J \end{array}$$
where *J* is the wavelet transformation level.

## 4. Experiment Results

### 4.1. Perceptual Security

Images and videos are tightly related to perceptual quality. Generally, social image encryption should not only be secure when they are communicated on social media platforms but also secure in visual perceptual quality. The encrypted social image should be unintelligible for confidentiality. For the purpose of protecting them from unauthorized access through obscuring (encrypting) them, the social image is considered to be secure if its encrypted form is not accessed by unauthorized users. To limit the illegal viewing of a social image, the more degraded their perceptual quality is, the more secure the encryption algorithm is.

The fingerprint code segments are embedded in the TSH domain. The LL coefficients in the TSH wavelet transform domain are permuted via the jiugongge game. Then, the scrambled coefficients are compressed by JPEG2000 coding and encrypted using the probabilistic homomorphic cryptosystem. For perceptual security, the fingerprint information hidden in the encrypted and fingerprinted copy should not be perceptible or detectable. Figure 6 shows the related experimental results. Figure 6d shows that the quality of the decrypted and fingerprinted image does not have any apparent change. On the other hand, for the security of communication and storage, the encrypted image should not be intelligible. The visual quality of the encrypted image is shown in Figure 7b,c. In comparison with the decrypted–fingerprinted image or original image, the encrypted images become noise-like and are not actually perceptual. Thus, the proposed encryption algorithm is very effective to keep confidentiality.

### 4.2. Fingerprinting Performance Analysis

The fingerprint code segments are embedded into the social image in the compressed and encrypted domain. To keep the visual quality of fingerprinted images, the fingerprint codeword in the decrypted–fingerprinted copy should not be perceptible and perceptually undetectable. Figure 6d shows the decrypted–fingerprinted Lena image; it is very apparent that the fingerprinted image looks like the original image shown in Figure 6a. Multilevel fingerprint code segment embedding in parallel in the compressed–encrypted domain directly decreases the computational complexity. Furthermore, the multilevel community code segments are not embedded repeatedly. Most important of all, it does not require decompression or decryption operation to save computing time. The redistribution tracing algorithm may narrow down the content tracing within a community. It is unnecessary to search for all users in the social media platform.

In the proposed fingerprinting scheme, the time taken for fingerprint embedding is calculated and shown in Table 1. The fingerprint embedding process is performed on a Pentium(R) Dual-Core E5700 computer, and the software platform is MATLAB 9. According to Table 1, it is known that 1-level means the first-level wavelet transform, as do 2-level and 3-level. So, it can be seen that the proposed fingerprinting scheme is efficient. With the proposed operation, privacy protection services can be provided.

### 4.3. The Encryption Process

The jiugongge permutation only increases the unintelligibility of the encrypted social image. Although single coefficient permutation in all subbands via the jiugongge game can achieve a better effect than 8×8 blocks permutation in the LL subband, the former will take 64 times as much time as the latter will take. The latter can obtain almost the same encryption visual effect as the single coefficient permutation when the block size is 8×8. Furthermore, block permutation via the jiugongge game only took 1/16 of the time that single coefficient permutation took. Therefore, 4×4 blocks permutation via the jiugongge game can have better performance than the others.

On the other hand, even if the homomorphic encryption of the compressed bytes stream is cracked, the illegal user still cannot decrypt the social image because the jiugongge game key of the permutation in coefficient block encryption remains secret.

### 4.4. Ability of Resisting Exhaustive Attack

A good multimedia content encryption scheme should be sensitive to the keys. The key space should be very large to resist brute-force attacks. The total key space includes initial values and control parameters of the confusion and diffusion processes. The proposed encryption method has the following secret keys: (1) initial values $x_{0}$ and $y_{0}$ of the logistic map and the PWLCM system, respectively; and (2) control parameters *u* and η of the logistic map and the PWLCM system. The sensitivity of keys $x_{0}$, $y_{0}$, *u*, and η is about $10^{-16}$. The total key is considered to be more than $10^{16\times 4}=10^{64}$. With the large key space, the brute-force attack can be resisted. Figure 6c is the decrypted image with the wrong keys which have only a different initial value. On the contrary, Figure 6d is the decrypted image with the correct keys.

### 4.5. Comparative Analysis

A comparative analysis of the proposed technique with the existing state of the art is presented in this subsection. The compared technique is a joint watermarking and encryption algorithm to protect medical images [25]. The authors have suggested the marriage of encryption and watermarking. However, the proposed algorithm had high time complexity for encryption because of the abundant data in the images. On the other hand, watermarking and encryption are conducted in the JPEG-compressed domain. The approach is inefficient for social image distribution because it took the approach a large amount of time to distribute the common content. The proposed distribution scheme can overcome the aforementioned weaknesses in [25] by incorporating a chaotic map with the jiugongge game and the probabilistic homomorphic cryptosystem in the JPEG2000-compressed domain.

The proposed technique is perceptually efficient because the encryption, watermarking, and distribution can be performed in parallel. Furthermore, the use of multicast and unicast for content dissemination can avoid big data problems on social media platforms. The use of the jiugongge game for block permutation before the JPEG2000-compressed process can provide another layer of security protection for social images. This proves an improvement in the proposed scheme for social media dissemination over the existing watermarking and encryption technique.

Table 2 summarizes the features of our approach and the existing joint multimedia fingerprinting and encryption schemes. The existing schemes only encrypt content in the spatial domain or the transform domain. Most multimedia contents are stored or communicated in the compressed domain, but all the above schemes were not applied to JPEG2000 image sharing on social media platforms.

### 4.6. Distribution Performance Analysis

The JFE scheme involves the jiugongge game, the probabilistic homomorphic cryptosystem, and JPEG2000 compression based on the TSH wavelet transform with the SNA technique. The JPEG2000 compression technique can provide a scalable approximation matrix, which contains the most important low-frequency information of a given original image. The SNA technique can determine how to construct the hierarchical fingerprint code and how to distribute the multimedia content. The proposed scheme is scalable because it can reduce the burden of the server by only sending the small-size approximation matrix and using the hierarchical community structure to support the multimedia content distribution process.

We analyze the distribution efficiency of a social image. If the fingerprinted copy for each user is different, it will use the worst-case naive unicast approach for content dissemination. Each user has their own unique private channel for transmission of their fingerprinted copy in the unicast mode. If there is only one public channel for transmitting multimedia content to $N_{u}$ users, then $N_{u}$ copies will occupy the channel $N_{u}$ times. In the proposed content dissemination scheme, the primary content with a unique user fingerprint code segment will be delivered to each user with a naive unicast approach. The supplementary content with community fingerprint code segments will be distributed to users in the same community via multicast. Therefore, the hybrid unicast and multicast mode is cheaper than only using the unicast mode for all fingerprinted content distribution. The fingerprinted multimedia distribution scheme is shown in Figure 5.

As shown in Figure 2, the size of the multimedia content in the 3rd-level LL subbands is significantly reduced. Assume the content dissemination efficiency of the social media platform *D* is measured by the following ratio equation.
(19)$$\eta_{D}=\frac{m_{D}}{m_{0}}$$
where $m_{D}$ is a proportional value which means the bandwidth used by the social media platform *D*. $m_{0}$ is another proportional value which represents the bandwidth used in the communication channel. $m_{0}$ is regarded as the number of unicast times the broadcast channel is used. $m_{D}$ is the number of times the communication channel is actually used by *D*. Therefore, if a transmission performance of D1 is more efficient than D2, there exists
(20)$$\eta_{D1}<\eta_{D2}$$

Consider the delivered social image with size M×N. After the 3rd-level TSH wavelet transform, for a special discrete wavelet transform, the LL subband is $\frac{M}{64}\times \frac{N}{64}$. All other subbands are $\frac{63M}{64}\times \frac{63N}{64}$. When traditional fingerprinted content dissemination is used, there will be fingerprinted content with size $M\times N\times N_{u}$ to be delivered.

However, once the proposed content distribution scheme is used, the content to deliver will be $\frac{M}{64}\times \frac{N}{64}\times N_{u}+\frac{63M}{64}\times \frac{63N}{64}\times C$. *C* is the number of communities. It is very apparent that the proposed content dissemination method will transmit less fingerprinted content on the social media platform than those existing methods.

Figure 5 depicts the proposed content distribution method, as well as its multimedia sharing social network that lies in the social media platform. The key idea is that each fingerprinted copy of a social image needs to be distributed to the corresponding user. For the common fingerprinted content, we first simply multicast the common fingerprinted content to all users in the same community. On the contrary, the unique fingerprinted content is small in volume, and it can be easily unicast to the corresponding user. At last, the proposed scheme can extend the distribution network to incorporate both the multicast and unicast channels. In this case, every user has two keys: a community key Kp is used to decrypt the common encrypted–fingerprinted data, and a unique key decrypts the encrypted–fingerprinted approximate subband.

## 5. Conclusions

The traditional joint encryption and watermarking scheme cannot be applied to secure multimedia dissemination for social media platforms because of their tremendous scale. In this work, the proposed social fingerprinting method in the compressed–encrypted domain is proposed to protect social image dissemination. The experiment results and performance analysis exhibit the security and effectiveness of the proposed scheme. The main objective of this research is to provide a useful synthesis of SNA for the field of secure multimedia dissemination for social media platforms. Although the proposed scheme has shown some promising results, there is still much work to be performed in the future. The limitation of the proposed scheme is not adapted to the dynamic property of social networks on the social media platform. For future work, we will refine the study of social image secure dissemination according to the dynamic change in social networks.

## Figures and Tables

**Figure 1 entropy-25-01031-f001:**
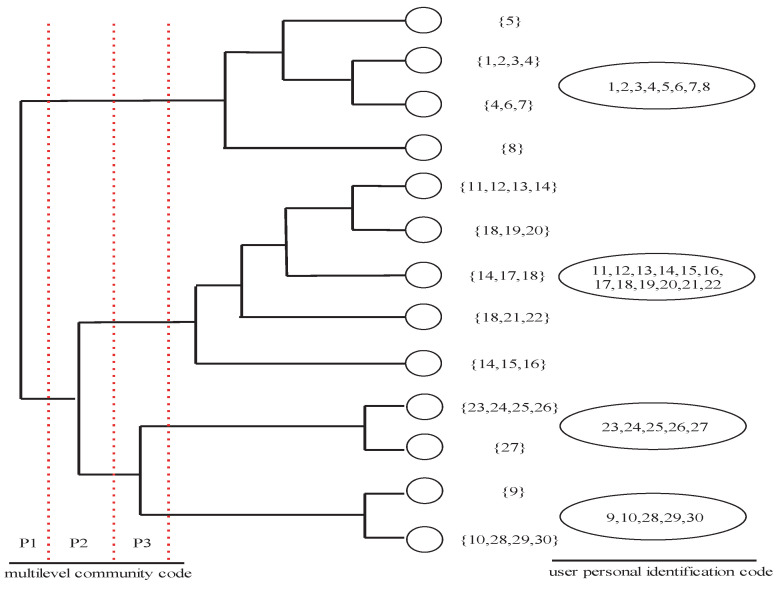
Encoding using SNA.

**Figure 2 entropy-25-01031-f002:**
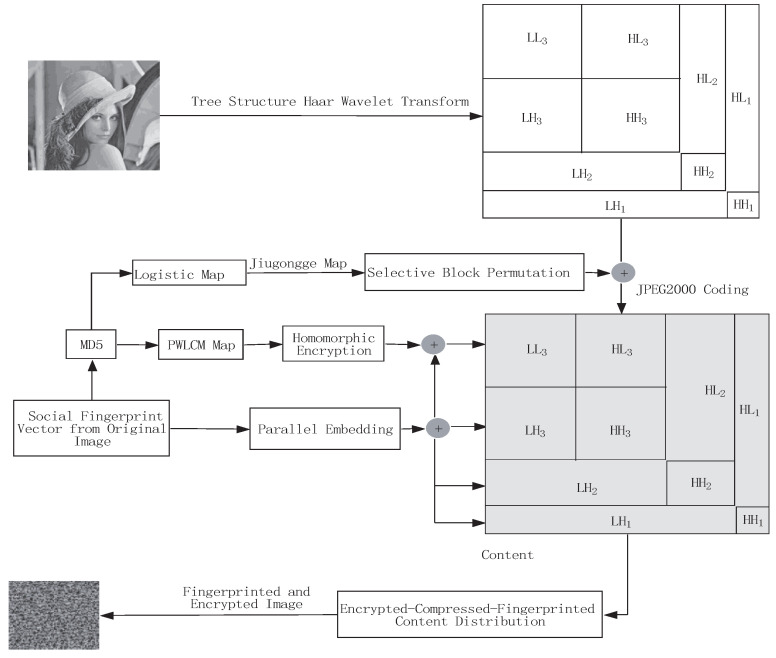
The proposed social image protection scheme.

**Figure 3 entropy-25-01031-f003:**
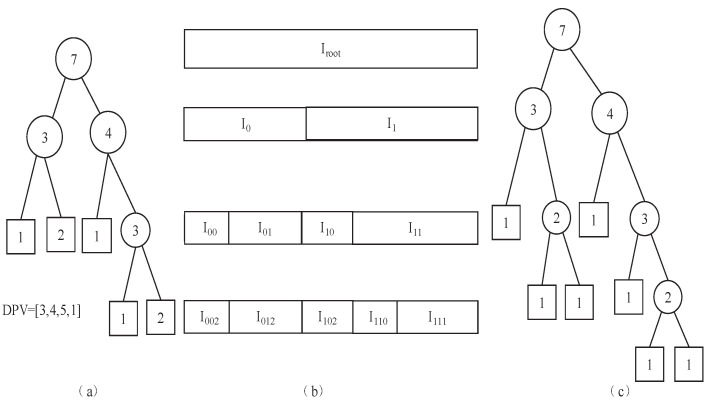
DPV production scheme. (**a**) Tag binary tree, (**b**) the corresponding interval, (**c**) the corresponding complete binary tree.

**Figure 4 entropy-25-01031-f004:**
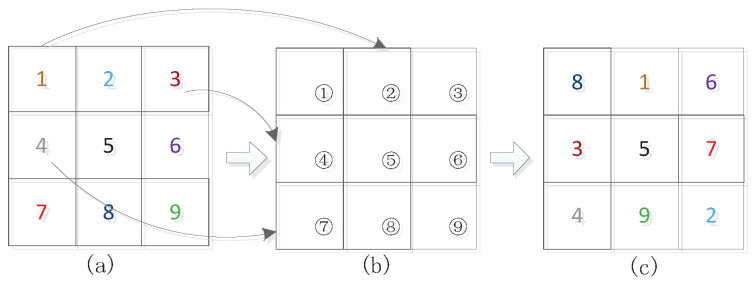
Jiugongge block permutation. (**a**) Original block layout, (**b**) Nine palace map scrambling rules, (**c**) Rearranged block layout.

**Figure 5 entropy-25-01031-f005:**
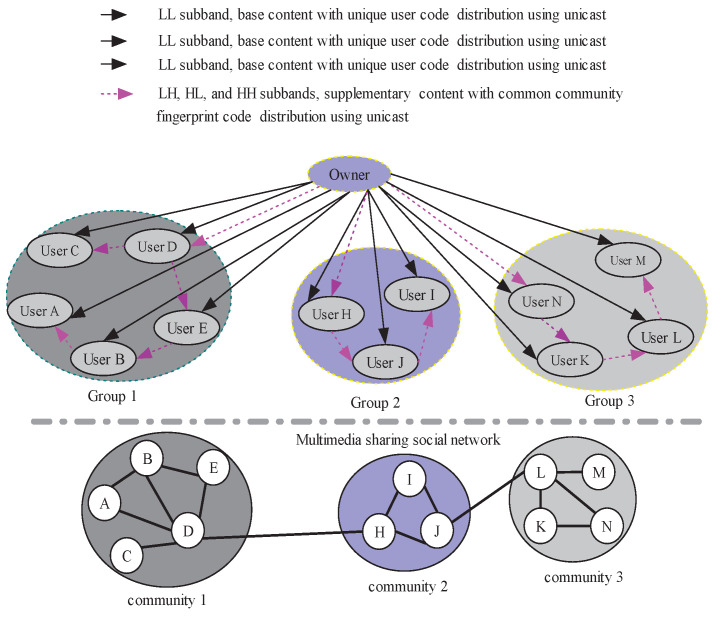
Social image distribution.

**Figure 6 entropy-25-01031-f006:**
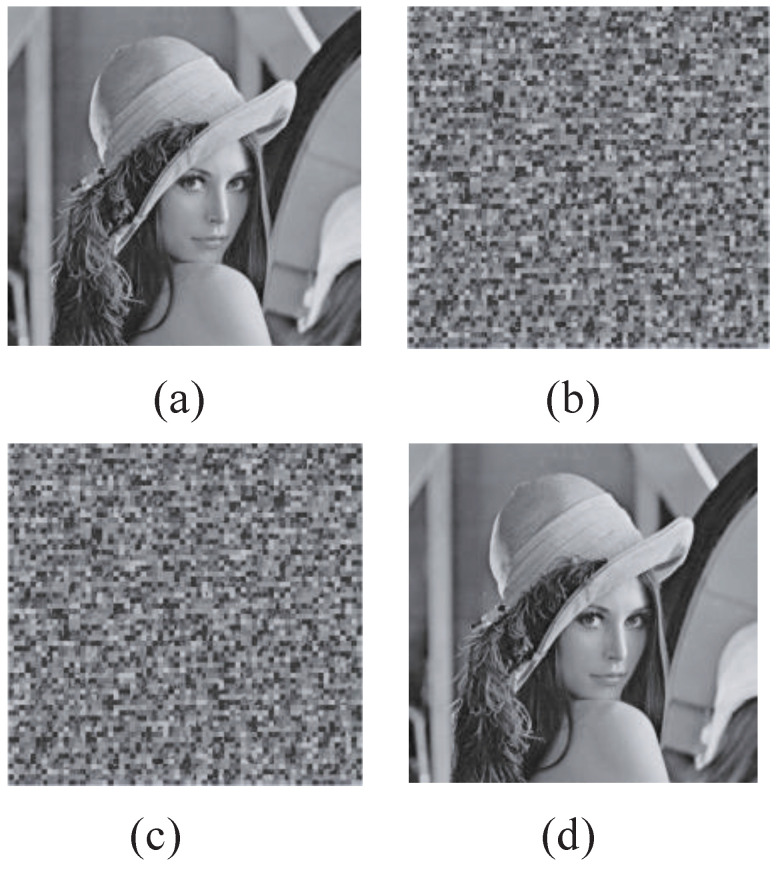
Experimental results. (**a**) The original image. (**b**) The encrypted image. (**c**) The decrypted image with a different initial value. (**d**) The decrypted image with correct parameters.

**Figure 7 entropy-25-01031-f007:**
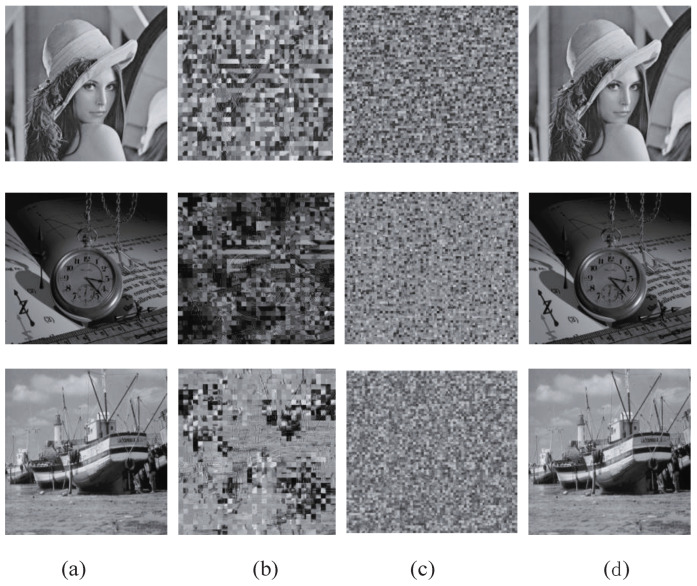
Evaluation of the encryption process: (**a**) original images, (**b**) 8×8 blocks permutation via jiugongge game and homomorphic encryption of compressed bytes stream in the LL subband, (**c**) images which are permutated by 4×4 blocks permutation via jiugongge game, (**d**) decrypted images with social fingerprints.

**Table 1 entropy-25-01031-t001:** Fingerprint embedding cost (ms).

Image	1-Level	2-Level	3-Level
Lena	26	19	13
Watch	27	19	12
Boat	26	18	12

**Table 2 entropy-25-01031-t002:** Examples of JFE schemes.

	Scheme	Content	Domain	Fingerprints	For SN
Ours	Jiugongge game, chaotic maps	Part	Compressed	Coding using SNA	Yes
[51]	RC45	Full	Spatial	Unknowable	No
[52]	Replacement, chaotic maps	Full	Quantization	Unknowable	No
[53]	Stream cipher	Full	Compression	Unknowable	No
[54]	EIGamal cryptosystem	Full	Spatial	Unknowable	No
[25]	AES	Full	JPEG-LS	Unknowable	No
[33]	RSA	Part	TSH	Unknowable	No

## Data Availability

Not applicable.

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
