# Peer review of "Hybridized Hierarchical Watermarking and Selective Encryption for Social Image Security"

_entropy, 2023, doi:10.3390/e25071031_

Round 1

Reviewer 1 Report

The article presents an encryption/watermarking method for securing social images. Some commens/suggestion follow:

- Section1:

   * Line 66. "by the cryptosystems [11-15]. But .." should be "by cryptosystems [11-15], but .."

 * Line 100: SNA acronym should be defined although it has been previously defined in Abstract, and a reference should be added.

- Section 2.

   * A brief paragraph introducing the section would be interesting

  * Section 2.3. Authors should carefully justify why they are using MD5 as cryptographic function. It presents some weaknesses, and it is not used anymore in cryptographic standards [ref1][ref2]

   * Phrase "If the link exists, the edge aij (i, j=1, . . . , M) is equal to 1, or zero." is incomplete.

* Line 166: Acronym PWLCM should be defined. References should be added for PWLCM and logistic chaotic maps.

 *Line 177. A reference for JFE scheme should be added.

* Line 196: Please, add some reference for the "jiugongge game".

* Line 249: Please, define BS acronym.

* Line 269. Phrase "Discontinuity Point Vector (DPV) can define every TSH wavelet basis for a given N, splitting scheme of the TSH transform can be decided by the DPV" needs to be rewritten. Additionally, subsection 3.2 should be extended in order to explain more in detail the role of DPV, and the corresponding references should be added.

- The use of iugongge game should be justified, detailing what advantages presents against other permutation options.

- Section 3.5. Proposed cryptosystem requires a security analysis or a reference including such analysis.

 - A flow diagram of the complete protection process would be interesting for the reader.

- There are several typos and grammatical errors in the manuscript, it should be thoroughly revised.

- Section 4.2. Performance analysis should include numerical results and the hardware/software platform used for the experiments.

References:

[ref1] Stevens, Marc, Arjen K. Lenstra, and Benne De Weger. "Chosen-prefix collisions for MD5 and applications." International Journal of Applied Cryptography 2.4 (2012): 322-359.

[ref2] Liang, Jie, and Xue-Jia Lai. "Improved collision attack on hash function MD5." Journal of Computer Science and Technology 22.1 (2007): 79-87.

As commented previously, there are several typos and grammatical errors in the manuscript, it should be thoroughly revised. Additionaly, there are too many short sentences.

Reviewer 2 Report

Summary:

This paper addresses the importance of social image security due to the emergence of cloud computing and social media platforms. To improve the security of social images, the study proposes a method that uses hybrid hierarchical watermarking and selective encryption techniques.

Vulnerabilities:

A limitation of this study is that the proposed method may not adapt well to the dynamic nature of social media platforms. In addition, the study lacks a detailed analysis of the experimental results.

Strengths:

The hybrid hierarchical watermarking and selective encryption techniques proposed in this study are very beneficial for improving the security of social images. This technology prevents copyright infringement, protects against personal information leakage, and ensures data integrity.

Questions:

(1) The proposed method includes JPEG2000 image encryption and decryption and multimedia encryption using chaos maps, which are well-known techniques. However, these methods may not adapt well to the dynamic nature of social media platforms, so improvements may be needed.

What are possible improvements for this aspect?

(2) The paper lacks an objective performance evaluation. How can this be improved?

Reviewer 3 Report

This paper proposes a secure scheme for social image dissemination on social media platforms. The main objective is to make a map between the tree structure haar is (TSH) transform and the hierarchical community structure of a social network. 

- "The privacy protection method proposes how to use the JFE for secure dissemination in the compressed-encrypted domain.">> please explain how this can be achieved?

-"The proposed scheme can avoid big data problems on social media platforms to the utmost extent with the JFE in the compressed domain." >> please explain how this can be achieved?

-There is no threat model to describe the privacy threats and attacker capabilities.

-There is no security and privacy analysis to discuss how the proposed scheme achieves the contributions discussed.

-How is this work different than other works in literature. please explain

needs proofreading 

Round 2

Reviewer 1 Report

The manuscript has been significantly improved. Nevertheless, there are some typos that should be corrected:

   - Title of Section 2.3. It should be "Secure Hash Algorithm (SHA-3)"

  - Line 163: "A cryptographic hash function is can map.. " should be "A cryptographic hash function can map "

   There are still some typos in the manuscript (see comments to authors), a final review of English Language would be desirable.

Reviewer 3 Report

authors have implemented comments from previous round

authors have implemented comments from previous round

Author Response

Thank you for your positive comment.